# Increasing the Survival of a Neuronal Model of Alzheimer’s Disease Using Docosahexaenoic Acid, Restoring Endolysosomal Functioning by Modifying the Interactions between the Membrane Proteins C99 and Rab5

**DOI:** 10.3390/ijms25136816

**Published:** 2024-06-21

**Authors:** Maxime Vigier, Magalie Uriot, Fathia Djelti-Delbarba, Thomas Claudepierre, Aseel El Hajj, Frances T. Yen, Thierry Oster, Catherine Malaplate

**Affiliations:** 1Unité de Recherche Animal et Fonctionnalités des Produits Animaux (UR AFPA), Qualivie Project, UA 3998, USC INRAE 340, Campus INP, University of Lorraine, 54500 Vandœuvre-lès-Nancy, France; maxime.vigier2@gmail.com (M.V.); magalie.uriot@u-bordeaux.fr (M.U.); fathia.djelti@univ-lorraine.fr (F.D.-D.); thomas.claudepierre@univ-lorraine.fr (T.C.); catherine.malaplate-armand@univ-lorraine.fr (C.M.); 2Department of Biochemistry, Molecular Biology and Nutrition, Nancy University Hospital, 54000 Nancy, France

**Keywords:** docosahexaenoic acid, Alzheimer’s disease, APP, C99 protein, neuroprotection, membrane remodeling, endolysosomal system, protein interactions

## Abstract

Docosahexaenoic acid (DHA, C22:6 ω3) may be involved in various neuroprotective mechanisms that could prevent Alzheimer’s disease (AD). Its influence has still been little explored regarding the dysfunction of the endolysosomal pathway, known as an early key event in the physiopathological continuum triggering AD. This dysfunction could result from the accumulation of degradation products of the precursor protein of AD, in particular the C99 fragment, capable of interacting with endosomal proteins and thus contributing to altering this pathway from the early stages of AD. This study aims to evaluate whether neuroprotection mediated by DHA can also preserve the endolysosomal function. AD-typical endolysosomal abnormalities were recorded in differentiated human SH-SY5Y neuroblastoma cells expressing the Swedish form of human amyloid precursor protein. This altered phenotype included endosome enlargement, the reduced secretion of exosomes, and a higher level of apoptosis, which confirmed the relevance of the cellular model chosen for studying the associated deleterious mechanisms. Second, neuroprotection mediated by DHA was associated with a reduced interaction of C99 with the Rab5 GTPase, lower endosome size, restored exosome production, and reduced neuronal apoptosis. Our data reveal that DHA may influence protein localization and interactions in the neuronal membrane environment, thereby correcting the dysfunction of endocytosis and vesicular trafficking associated with AD.

## 1. Introduction

Docosahexaenoic acid (DHA, C22:6) is the most abundant omega-3 polyunsaturated fatty acid (PUFA) in brain phospholipids, particularly in the synaptic and intracellular membranes of neurons [1]. By influencing the physicochemical properties of the bilayer, DHA can modify the membrane architecture, with possible consequences on the organization and composition of liquid-ordered microdomains, i.e., lipid rafts, as well as on the interactions with and between associated proteins [2,3]. Therefore, DHA is most likely involved in physiological processes, such as membrane receptor responsiveness, signal transduction, endocytosis, and neurotransmitter release [4], making it crucial for all brain functions during development, as well as throughout life. By itself and/or its bioactive metabolites, DHA has also been widely reported to participate in several protective pathways against brain aging and cognitive loss [5]. Alterations in lipid metabolism have been associated with brain alterations in elderly or demented individuals [6], while DHA deficiencies have been detected in patients suffering from neurodegenerative diseases [7,8]. Sufficient levels of DHA in neuronal membranes therefore appear to need to be maintained in order to preserve brain functions.

Alzheimer’s disease (AD) is a neurodegenerative disease of the aging population, the prevalence of which is increasing due to a higher life expectancy. Understanding the causal mechanisms of AD is necessary to develop new therapies and strategies aimed at preventing or delaying it. Although AD is now considered multifactorial, amyloid precursor protein (APP) cleavage products, notably amyloid-β peptide (Aβ), are critical factors involved early in the series of neurotoxic mechanisms associated with the pathogenesis of AD [9,10,11,12]. The processing of APP occurs primarily in endosomes, a compartment that provides optimal conditions for secretase activity [13,14], and several genes influencing APP endocytosis, such as ApoE4, SorL1, Bin-1, or PICALM, have been identified as risk factors for AD [15,16,17].

The regulated recycling of membrane components, proteins, and lipids, is a crucial process for the maintenance of homeostasis and cellular functions, especially for neurons for which the post-mitotic status requires maintenance that is decisive for their survival. This process mainly involves vesicular trafficking carried out by the endolysosomal system consisting of a dynamic network of vesicles functionally linked to the plasma membrane. It is made up of a series of specialized organelles, including early endosomes, recycling endosomes, late endosomes, and lysosomes. The progressive maturation and trafficking of endosomes rely on morphological changes of the vesicles allowing the sorting and targeted degradation of proteins, under the control of small GTPases of the Rab family [18]. Also considered specific markers of the different vesicular compartments, Rab proteins are important membrane association molecules that function as molecular switches for which the active conformational state is linked to bound GTP. Three different sorting systems operate in these processes: targeted degradation, retrograde retrieval, and the secretion of exosomes, i.e., extracellular vesicles for which a shuttling role has been associated with various physiological functions, as well as the propagation of pathogenic molecules [19], including cytotoxic products linked to AD [20].

The endolysosomal system appears to be an actor of which failure is central in a multitude of neurodegenerative diseases, suggesting the potential interest in this pathway for new therapeutic strategies. Dysfunctions of this system, such as hypertrophy of endosomes and blockage of vesicular trafficking, are recognized early cellular hallmarks of AD [10]. These dysfunctions were observed in neurons generated from induced pluripotent stem cells from individuals with familial and sporadic AD [17] and from Down syndrome fibroblasts [21]. Thus, the enlargement and aberrant signaling of endosomes are the immediate consequences of the pathological overactivation of Rab5 [21,22,23]. A link between endolysosomal dysfunction and the intraneuronal accumulation of the C-terminal fragment of APP produced by β-secretase (β-CTF or C99) has been demonstrated in vivo [24,25]. The association of C99 with endosomal enlargement was also confirmed in neurons of AD patients [22]. Mechanistically, C99 produced in the endosomal membrane can recruit the signaling molecule APPL1, which stabilizes the active form of Rab5 (GTP-Rab5) of early endosomes [22,23], thereby contributing to endosome enlargement and blocking their maturation.

In previous work, we demonstrated that the membrane enrichment of DHA can induce membrane remodeling that allows, in particular, the colocalization of ciliary neurotrophic factor (CNTF) receptor subunits in rafts, which then promotes the availability of this receptor for its ligand, as well as the functionality of the associated neuroprotective signaling pathway [26,27]. Therefore, based on its effect on membrane architecture and organization, we hypothesized that DHA might also ameliorate endolysosomal dysfunctions induced by APP cleavage products. Here, we investigated the effect of DHA supplementation on endosomal trafficking and exosomal production in an AD cell model, the human neuroblastoma SH-SY5Y cell line expressing recombinant human APP protein carrying the Swedish double-mutation K670N/M671L (APPswe), which promotes the amyloidogenic processing of APP [28].

## 2. Results

### 2.1. Characterization of the Endolysosomal Phenotype of Wild-Type or Overexpressing APPswe Model Cell Lines

With the aim of studying how the neurotoxic properties of the Aβ peptide can impact the functioning of vesicular trafficking, we developed a first series of experiments on cells from the wild-type SH-SY5Y cell line and the derived line expressing APPswe and accepted as a cellular model of AD because it develops a cholinergic phenotype after differentiation with retinoic acid. These two lines exhibit very different levels of APP and cleavage products, providing the opportunity to study the effect of continued exposure to the corresponding neurotoxic agents.

We first studied some differences related to the overexpression of APPswe, notably the endosomal phenotype of each of the two retinoic acid-differentiated lines. Cultured cells were either harvested for protein extraction and Western blot analysis or fixed in paraformaldehyde for immunostaining. As the Swedish mutation of APP is well known to induce early endosome enlargement [29], we focused our attention on studying the early endosomal EEA1 protein as a marker. Immunofluorescence results (Figure 1a,b) in APPswe cells revealed a significant decrease in the number of early endosomes per cell (−42%, *p* < 0.05) associated with an enlargement of these vesicles (+23%, *p* < 0.05), both hallmark signs of blocking the endosomal vesicle trafficking [30].

As exosomal production is directly related, downstream, to the endosomal system, we also explored the production of exosomes by selectively quantifying the nanoparticles of a 50–120-nm diameter released into the medium using NTA. The exosomal nature of these particles was assessed by showing the presence of the specific marker ALIX, while their membrane nature was confirmed by the presence of the raft marker Flotillin-1 in the homogenates prepared from the exosome pellets. Our results (Figure 1c) showed a 2-fold lower production of exosomes from APPswe cells than from WT cells (*p* < 0.05). As expected, APP protein was detected in the exosomes produced by SH-SY5Y APPswe cells but neither in WT cells nor in the exosomes they produced (Figure 1d). However, immunoblot analyses were unable to reveal the presence of APP fragments (C99, C83, Aβ peptides) in these extracellular vesicles, likely due to their low amounts in these samples.

### 2.2. Effects of APPswe Overexpression on Death/Survival Balance in SH-SY5Y Cells

The alterations to the endolysosomal pathway, as observed in SH-SY5Y APPswe cells, are expected to be disabling. Different possible consequences were investigated under standard culture conditions. Apoptosis has been proposed as a possible mode of neuronal death in AD based on the presence of fragmented DNA [31,32]. Pyknotic neurons have also been found in adult PS/APP mice, exhibiting apoptotic changes, including DNA fragmentation and caspase-3 activation [33]. Therefore, we evaluated apoptosis in both cell lines using DAPI staining and the Western blot analysis of caspase 3 and Akt. As compared to WT cells, APPswe cells exhibited a 3-fold increase in the proportion of pyknotic nuclei (Figure 2a), a 5-fold increase in the cleaved caspase 3/pro-caspase 3 ratio (Figure 2b), and an 8-fold increase in the p-Akt/Akt ratio (Figure 2c). These results suggest that higher endogenous cytotoxicity and activation of the pro-apoptotic cascade could be possible consequences of the endolysosomal defects shown in Figure 1.

### 2.3. Involvement of APP Fragments in the Abnormalities Observed in SH-SY5Y APPswe Cells

We next aimed to determine whether these adverse effects could result from APP processing and the production of cleavage products, particularly the Aβ peptide for which neurotoxicity upon oligomerization has been abundantly documented (see [34] for review). To investigate this, SH-SY5Y APPswe cells were treated with 0.5 µM LY, a γ-secretase inhibitor, in order to inhibit Aβ peptide production. This dose was selected using MTT and LDH assays that allowed us to define the highest non-toxic dose to inhibit the production of the amyloid peptide. We verified the decrease in Aβ-40 and Aβ-42 peptides released into the culture medium (Figure 3a), confirming the effective inhibition of γ-secretase in these cells. As expected, the significant accumulation of C99 fragments (+146%, *p* < 0.01) was concomitantly observed, shown based on Western blot results (Figure 3b).

Considering that the endogenous cytotoxicity observed in SH-SY5Y APPswe cells is presumably associated with Aβ production, LY was expected to prevent the cytotoxic mechanisms. On the contrary, the apoptosis of LY-treated SH-SY5Y APPswe cells actually increased (Figure 3c), with a significantly higher percentage of pyknotic nuclei (+21%, *p* < 0.01). This was accompanied by an even greater reduction in exosome production (−71%, *p* < 0.01) (Figure 3d) upon γ-secretase inhibition. In addition, even if the quantification was difficult because of the superposition of the objects to be counted or measured, the number of enlarged EEA1-positive endosomes appeared to be increased in LY-treated SH-SY5Y APPswe cells (Figure 3e), suggesting that C99 accumulation could also worsen the defects of the endosomal pathway. This was further confirmed via Western blot analysis (Figure 3f), which revealed significantly higher levels of EEA1 (+105%, *p* < 0.05) in LY-treated SH-SY5Y APPswe cells.

### 2.4. Effects of DHA on Abnormalities in SH-SY5Y-APPswe Cells

Next, these cells were treated with 0.5 µM DHA for 48 h, under the same conditions used previously to demonstrate its neuroprotective effects [27,35]. Western blot results (Figure 4a) revealed a significant decrease in EEA1 protein levels after DHA treatment (−47%, *p* < 0.05). Immunofluorescence analysis (Figure 4b) revealed a diminution of early endosome enlargement in DHA-treated SH-SY5Y APPswe cells, suggesting that blocking vesicle trafficking was mitigated by DHA. Interestingly, exosome production (Figure 4c) by these DHA-treated cells was significantly higher (+133%, *p* < 0.01). These latter results clearly indicate that DHA can improve the defects and suppress the blocking of the endolysosomal system in the SH-SY5Y APPswe cells. This pathway could therefore represent a potent new neuroprotective mechanism activated by DHA, which was confirmed by the significant decrease (−32%, *p* < 0.001) in the ratio of pyknotic nuclei in SH-SY5Y APPswe cells observed after DAPI staining (Figure 4d).

Since we demonstrated a correlation between the defects and cytotoxic effects with the accumulation of C99 in the SH-SY5Y APPswe cells, it was then necessary to study whether DHA neuroprotective properties could involve a change in APP processing and C99 production in these cells. Western blot analysis showed no effect on C99 levels after DHA treatment (Figure 4e), suggesting that the neuroprotection observed in the SH-SY5Y APPswe cells could not be explained by a decrease in the production of C99 from APP.

### 2.5. Involvement of C99–Rab5 Interaction in Endosomal Pathway Alterations and Remediation Mediated by DHA

We tracked the C99–Rab5 interactions in SH-SY5Y APPswe cells treated with LY or DHA by using the Duolink^®^ colocation staining assay. Our results (Figure 5a) showed that γ-secretase inhibition and subsequent C99 accumulation increased the size, as well as the number, of C99–Rab5 colocation spots, without any change in Rab5 levels (Figure 5b), thereby emphasizing the link of this interaction with the endolysosomal defects. On the contrary, DHA significantly reduced the colocation spots in terms of size and in number, modifying neither Rab5 (Figure 5b) nor C99 levels (Figure 4e). This suggests that membrane organization may have been remodeled upon DHA incorporation in a manner that disfavors deleterious interactions between these two proteins in the endosomal membrane. This remodeling could then be decisive in lifting the sequestration of vesicles in the form of early endosomes and therefore in releasing their maturation and trafficking.

## 3. Discussion

The work presented in this article characterizes a new mechanism to which DHA could contribute to preserve the survival of neurons exposed to amyloid stress. Thanks to its ability to influence the microarchitecture and membrane organization, we provide here proof that even in cells continuously subjected to APP processing and the subsequent production of neurotoxic products, DHA seems to be capable of preventing the interaction between C99 and Rab5 proteins (Figure 5), thus improving the deleterious blockage of vesicular trafficking found in different models of AD as in the brains of patients.

Endolysosomal abnormalities in neurons are among the earliest events described in AD [36], as well as in a wide range of neurodegenerative diseases [37]. As amyloidogenic APP metabolism occurs preferentially in early endosomes [38], the blockage of their trafficking and addressing pathways from this compartment appears to be a key event that promotes the production of toxic amyloid fragments from APP proteolysis. In this work, we have shown that due to the unregulated overexpression of APP, cells of the recombinant SH-SY5Y-APPswe line are continuously exposed to these fragments, which subsequently generates chronic neurotoxic effects, such as the blockage of the endolysosomal vesicle trafficking and the decrease in exosome production (Figure 1), associated with pro-apoptotic conditions (Figure 2). Experiments designed to inhibit γ-secretase to avoid Aβ production and supposedly related neurotoxicity led to the opposite effects, since the endolysosomal abnormalities were further exacerbated by the accumulation of the C99 fragment (Figure 3). In contrast, these abnormalities were all corrected in the SH-SY5Y-APPswe cells cultured in media supplemented with DHA, though C99 production was unchanged (Figure 4).

Thus, in SH-SY5Y-APPswe cells, we observed an enlargement of early endosomes, as well as a decrease in lysosome-specific LAMP1 staining and in LAMP1 protein levels, comparably to those previously described in other cell models overexpressing APP [24,30]. This agreement with the literature confirms that APP and its cleavage products, through the amyloidogenic pathway, are associated with endosome abnormalities in AD brain cells and in various models. For example, hypertrophied EEA1-positive endosomes have also been found in blood mononuclear cells and fibroblasts from patients with mild cognitive impairment or AD [29,39], as well as in mouse model brains and cultured neurons [24,30]. Also, the lower number of LAMP1-positive structures and lower LAMP1 protein levels could reflect the inability of endosomes to evolve to the lysosome stage, leading to a defect in the degradation of obsolete or toxic components, as described in the same model [40]. In the literature, several studies have observed elevated levels of proteins involved in intracellular trafficking, including EEA1 and LAMP1, in the postmortem brains and CSF of AD patients [41,42,43]. These observations thus clearly position endolysosomal abnormalities among the pathophysiological mechanisms involved in AD. Interestingly, levels of endocytosis-related proteins (clathrin, dynamin, Rab5, caveolin-1) measured in the human brain correlate with age but not with levels of Aβ40 and Aβ42 peptides [44]. These results suggest that these abnormalities may emerge with age and most certainly under the influence of other AD risk factors, promoting APP endocytosis and its subsequent cleavage by endosomal BACE-1, which initiates the amyloid cascade. This suggests that endolysosomal dysfunctions, such as those we found in SH-SY5Y-APPswe cells, may be involved early in AD pathogenesis, as previously proposed [45]. By promoting amyloidogenic APP proteolysis, the blockage of endolysosomal trafficking could therefore be a cause of the accumulation of amyloid fragments, rather than a consequence of their production and exposure. The study of the possible influence of AD risk factors on endocytosis and vesicular trafficking processes would therefore be of particular interest to assess this hypothesis.

We also show, in this manuscript, that the SH-SY5Y-APPswe cells produce fewer exosomes than wild-type cells, suggesting that the blockage of vesicular trafficking at the early endosome stage may be associated with a decrease in the exosome secretory capacity. Endosomal abnormalities and autophagy defects have been widely described in the literature in different AD models [46]. However, limited data are finally available on exosome production in the continuum of the endolysosomal system, especially in models overexpressing APP. Exosome production represents another exit pathway from endosomes. Exosomes are described as transport forms of neurotoxic oligomeric forms of the Aβ peptide and have been proposed to contribute to its intercellular spread in the brain [14,47]. These authors consider exosomes as a pathway for the clearance and elimination of toxic products, such as the amyloidogenic cleavage products of APP, by the cell. One study also showed that Aβ-containing exosomes are neurotoxic when brought into contact with primary neurons in culture, conferring on these extracellular vesicles the function of a vector for the dissemination of neurotoxic agents, at least in vitro [48]. Therefore, these two aspects could reflect a common mechanism, one being the consequence of the other, as the production of exosomes intended to protect the producing cell leads to the spread of their enclosed deleterious compounds to the other cells. Our results show that the SH-SY5Y-APPswe cells produce fewer exosomes than wild-type cells, suggesting that the removal of toxic amyloid products is impaired. Blockage of the endosomal trafficking could therefore not only promote the amyloidogenic pathway and the production of toxic APP fragments but also participate in their intra-endosomal accumulation following the inhibition of their maturation to lysosomes. Such accumulation could contribute to an auto-intoxication of SH-SY5Y-APPswe cells, a process that would then participate in the neurodegenerative processes associated with AD. Limited data only are available in the literature on the link between APP expression and the capacity to produce cellular exosomes. In contrast to our results, no difference in the levels of exosomes produced was reported in the APPswe-overexpressing murine neuroblastoma line N2a as compared to the wild-type [47], but no information was provided on the status of the endolysosomal system in these cells to allow for an explanation of these differences.

Due to the dysfunction and damage described above and observed in the SH-SY5Y cells that overexpress APPswe, cell survival was observed to be continuously threatened. When γ secretase activity and subsequent Aβ peptide production were inhibited in these cells, we observed that the cells were not protected from apoptosis as could be expected based on the established Aβ neurotoxic properties. On the contrary, the dysfunctions were major, with an aggravation of the early endosome abnormal phenotype and of the endolysosomal trafficking blockage, thereby promoting stressful conditions and cell death.

The neurotoxicity of the Aβ peptide has been widely demonstrated in the literature and obviously cannot be questioned, even though it certainly imposes conformational changes and oligomerization to the monomeric peptide prior to acquiring its cytotoxic properties [49,50]. However, at a time when the relevance of the amyloid cascade is being questioned and in the face of the modest results obtained from clinical trials targeting Aβ, our results shed new light on the importance of attributing primary responsibility to the C99 fragment and later involvement of the Aβ peptide in the observed neurotoxicity. We hypothesized that C99 production could initiate early dysfunction of the producing neuron, while Aβ could amplify the neurotoxic response by dispersing it to the neighboring cells. The deleterious consequences of the accumulation of C99 fragments in neurons have already been observed. Studies in animal models of AD have provided evidence of neuroinflammation, i.e., microglial activation and astrogliosis, after the accumulation of C99 fragments, independent of Aβ levels [24,51]. In the AD brain, C99 fragments were detected specifically in vulnerable neurons undergoing neurodegeneration and at levels correlated with the severity of cognitive impairment, whereas Aβ peptides are found in all neurons, suggesting that C99 is more responsible for neuronal cell death than Aβ [9].

Several mechanisms have been proposed as hypotheses to explain the abnormalities of the endolysosomal system, such as those described in this study. One of them focused on the influence of membrane asymmetry on these phenomena. This asymmetry, which is essential to the functioning of the cell, could be modified by C99 as a result of its effect on the activity of a lipid flippase [52]. This recent discovery strongly suggests that membrane organization and lipid composition may play a key role in these mechanisms.

Other hypotheses proposed that C99 accumulation facilitates an interaction with the early endosome-specific Rab5 GTPase and the formation of complexes also involving the adaptor protein APLP1. As a result, Rab5 is hyperactivated in its GTP-bound form, leading to the blockage of vesicular trafficking at the early endosome stage [22,53] and to endolysosomal dysfunctions, such as those observed in SH-SY5Y-APPswe cells [53,54]. Furthermore, abnormal Rab5 hyperactivation has been observed in AD patients and in mouse models [37], inducing a blockage at the same stage of the early endosome associated with various prodromal and degenerative features of AD [23,44,55,56]. Our work also advocates for the importance of protein interactions involved in blocking endosomal trafficking in APPswe cells, in particular those that allow C99 to sequester the hyperactivated Rab5 at the membrane of early endosomes, thus preventing vesicle maturation and progression through the endolysosomal system and towards exosome secretion. Wild-type SH-SY5Y cells only express the APP protein at a very low level. They therefore do not present detectable C99 protein and benefit from functional vesicular trafficking. On the other hand, the blockage of this traffic observed in APPswe cells can be explained by the production of C99, which promotes deleterious interactions with Rab5. It is indeed the levels of C99 that seem to be decisive in this neurotoxic mechanism, since the inhibition of γ-secretase activity by LY worsens the situation without increasing the levels of Rab5, but only those of C99 (Figure 5).

Since C99 and Rab5 interactions take place at the endosomal membrane, we hypothesized that the environment and spatial organization of the lipid bilayer should favor the conditions for these interactions and more generally could determine most intermolecular contacts in the cell membranes. The neuroprotective potential of DHA has been well documented in numerous observational and basic studies, as reviewed [5,57]. We have previously demonstrated the neuroprotective capabilities of DHA, including the stimulation of anti-apoptotic and pro-survival pathways against soluble oligomeric Aβ-induced neurotoxicity [35], prevention of age-related cognitive decline, and the limitation of age-related membrane remodeling [27]. Based on our previous results on the neuroprotective properties of DHA, it was therefore interesting to assess whether the beneficial action of this *n*-3 PUFA on the neuronal membrane and cell viability could also result from promoting functional vesicular trafficking in the endolysosomal system. Studies in the literature have reported that DHA may influence endosomal functions [58,59,60], based on the premise that endocytosis and subsequent vesicle trafficking steps are primarily influenced by the physical properties and the spatial organization of the membrane, its lipid composition, and protein interactions and activities [52]. DHA is known to stimulate the synthesis of phosphatidylserine, the presence of which in the inner leaflet of neuronal membranes contributes mainly to membrane asymmetry [57]. This asymmetry of endosomal membranes determines the ability of DHA to trigger membrane budding and subsequent vesicle trafficking [52]. DHA has also been shown to promote endocytosis and optimal endosomal trafficking by interacting with clathrin and specific Rab proteins [61,62]. Regarding its impact on membrane remodeling, in vitro and in vivo evidence has revealed that DHA influences the lateral segregation and location of membrane-associated proteins [36,63]. The results presented here show that DHA can remove the blockage of endosomal trafficking and reactivate exosome production in SH-SY5Y-APPswe cells, thereby decreasing neuronal cell death. We clearly demonstrate that the treatment of these cells with DHA reduced the possibility for these two proteins interacting with each other without affecting their respective levels. Therefore, it seems reasonable to hypothesize that the lipid composition and microarchitecture of the membranes were modified in the SH-SY5Y-APPswe cells treated with DHA, which may have reorganized the raft domains so that C99 and Rab5 are kept away from each other. This hypothesis completely agrees with the effect of DHA on the subunits of the CNTF receptor [27], which points to the determinant influence of this PUFA on membrane features and the segregation of associated proteins. Given its diverse neuroprotective properties, we propose that DHA may prevent the endogenous cytotoxicity observed in SH SY5Y-APPswe cells by preserving the function of the endolysosomal pathway impaired in early AD, very likely through membrane remodeling. This hypothesis now requires further confirmation, which we can also consider use of the SH-SY5Y cell differentiation protocol very recently published by D’Aiola et al. [64] in order to have a human model of the 3D culture of functional cholinergic neurons, offering an ethical alternative to animal experimentation in compliance with the 3 R principles.

## 4. Materials and Methods

All high-grade chemical and reagents were purchased from Sigma Aldrich (L’lsle-d’Abeau Chesnes, France) unless otherwise specified.

### 4.1. Cell Culture, Neuronal Differentiation, and Treatments

SH-SY5Y cell lines were a kind gift from Dr Marie-Claude Potier (ICM, Paris, France). Cells from both lines were cultured in Dulbecco’s modified Eagle’s medium (DMEM, Gibco, ThermoFisher Scientific, Illkirch-Graffenstaden, France) supplemented with 10% fetal bovine serum (FBS, Gibco), 10,000 IU/mL penicillin, and 10,000 IU/mL streptomycin and maintained at 37C in a humidified 5%-CO_2_ atmosphere. In all experiments, once confluence reached 60–70%, the cells were differentiated in the presence of 10 µM retinoic acid over 5 days in the Neurobasal medium (Gibco) supplemented with 3% FBS, 10,000 IU/mL penicillin, and 10,000 IU/mL streptomycin (Invitrogen, ThermoFisher Scientific, France). Cells were treated in serum-free neurobasal medium in the presence of LY450139 (LY) at 0.5 µM for 24 h or in the presence of DHA (Gibco) at 0.5 µM for 48 h. Cells were either recovered for Western blot analysis in RIPA lysis buffer (Sigma Aldrich) supplemented with 40 µg/mL cOmplete Protease Inhibitor Roche (Sigma Aldrich) or for immunostaining after fixation in 4% paraformaldehyde.

### 4.2. Western Blot Analysis

Protein sample preparation, SDS-PAGE, and immunoblotting were performed as previously described [33]. Briefly, cell proteins were resolved on 6 to 15% denaturing polyacrylamide gels prior to transfer onto polyvinylidene difluoride (PVDF) membranes. Membranes were activated in 100% methanol, then saturated in TBS with 5% skimmed milk or bovine serum albumin, and incubated overnight with specific antibodies. Primary antibodies of interest, i.e., rabbit monoclonal anti-Rab5 (35475, Cell Signaling Technology, Ozyme, France), mouse monoclonal 6E10 anti-APP/Aβ (BioLegend Europe, Amsterdam, The Netherlands), 82E1 anti-C99 (10323, IBL-America, Minneapolis, MN, USA), anti-EEA1 (48453, Cell Signaling Technology), and anti-β-tubulin (T5201, Sigma Aldrich) antibodies, were diluted for use at 1:1000. Peroxidase-conjugated secondary antibodies, i.e., goat anti-rabbit IgG, and horse anti-mouse IgG (70745 and 70765, respectively, Cell Signaling Technology), were diluted at 1:2000, and immunocomplexes were detected using the Supersignal chemiluminescence ECL kit (Merck Millipore, ThermoFisher Scientific, France). Gel images were obtained with the Fusion Imaging System (Fusion Fx5; Vilber Lourmat, Collégien, France). Apparent molecular masses were estimated based on protein standards (Prestained protein ladder plus, ThermoFisher Scientific, France). The densitometric analysis of western blots was performed using the image-processing ImageJ 1.52 software.

### 4.3. Immunocytochemistry and Quantification

Cells fixed in paraformaldehyde were immunostained with mouse anti-EEA1 antibodies, followed by Alexa Fluor-488 or Alexa Fluor-555 conjugated antibodies (Molecular Probes, Eugene, OR, USA), all diluted at 1:500. Nuclei staining analysis was performed using DAPI (1:10,000, Molecular Probes). Duolink proximity ligation assay (Duolink^®^ PLA, Sigma Aldrich) labeling was performed according to manufacturer’s instructions. The protocol was previously validated after verifying the absence of non-specific signals in different negative controls carried out with PLA probes used alone or with each antibody. Images were acquired randomly using an inverted confocal microscope (Fluoview10, Olympus, Rungis, France) and a 60X objective. Identical settings (intensity, contrast, exposure, specificity) were used for every culture condition. A total of 6–8 images (about 50 cells each) were acquired for each condition (3 independent experiments). Quantification of the number of particles and/or their area was performed using ImageJ software. Automatic particle analysis requires a binary, black and white image. A threshold range is set automatically and identically for all images to distinguish interesting objects from the background. All pixels in the image with values below the threshold are converted to black and all pixels with values above the threshold converted to white, or vice versa.

### 4.4. Exosome Isolation and Quantitation

The culture medium was replaced with a serum-free medium 1 day prior to exosome isolation. The culture supernatants were collected and subjected to centrifugation at 4 °C, at 300× *g* for 10 min, in order to discard cells. The supernatant was then sequentially centrifuged at 2000× *g* for 10 min to remove dead cells and at 10,000× *g* for 30 min to remove cell debris. Then, the supernatant was collected and centrifuged at 100,000× *g* for 1 h. The pellet was recovered in sterile water, and exosomes were quantified via nanoparticle tracking analysis (NTA) in order to measure the mean sizes and concentrations of particles harvested. A NanoSight LM10 instrument (Malvern Panalytical, Palaiseau, France) equipped with a scientific CMOS sensor and a 488 nm laser was used. Samples were prepared by making dilutions between 1:10 and 1:1000 in particle-free PBS to an acceptable concentration, according to the manufacturer’s recommendations. For each biological replicate, readouts were performed in quintuplicate. The instrument laser chamber was cleaned thoroughly between each sample. The camera shutter speed was set at 30.0 ms. NTA 3.1 software was used to process the 30 s video files of particles in suspension. Exosomes were considered as nanoparticles with sizes between 50 and 120 nm.

### 4.5. Quantitation of Aβ

After treatments, the culture medium was replaced with a serum-free medium for 24 h in the absence or presence of 0.5 µM LY. The culture supernatants were collected in polypropylene tubes. Complete protease inhibitor cocktail was added at 40 µL/mL, and the mixture was subjected to centrifugation at 4 °C, 2000× *g* for 10 min, to remove any cells in the supernatant. The levels of recombinant Aβ peptides in conditioned media were measured using the selective solid-phase sandwich ELISA kits to detect human Aβ40 (residues 1 to 40) and Aβ42 (residues 1 to 42) full-length peptides (Invitrogen).

### 4.6. Statistical Analyses

Statistical analyses were performed with PRISM 9.3.0 software (GraphPad Software) by using a two-tailed Student’s *t*-test or a non-parametric two-tailed Mann–Whitney U test for pairwise comparisons. All data are expressed as the mean ± SEM.

## 5. Conclusions

Membrane maintenance is an essential process in neurons, obliged to recycle their protein and lipid constituents in order to maintain the optimal spatial organization, integrity, and functionality of this important cellular compartment throughout cellular life. Endosomal vesicular trafficking is a major pathway involved in this process and its deficiency appears to contribute very early to the pathophysiology of AD.

In this paper, we showed that the SH-SY5Y APPswe cell line differed from the wild-type based on the chronic exposure to C99 fragments derived from APP catalysis, resulting in ongoing neurotoxicity involving the blockade of endolysosomal vesicle trafficking and a decrease in exosome production associated with cell apoptosis. These abnormalities are further exacerbated by the inhibition of γ-secretase activity, suggesting a major role of C99 fragments in triggering the neurotoxic cascade leading to neuronal death. We also demonstrated that the interaction of C99 with Rab5 is a key step in this deleterious cascade, since by preventing this harmful interaction, DHA was able to restore vesicular trafficking and exosome secretion, thus reducing cell death.

The ability to maintain a membrane organization adapted to associated functions could therefore be one of the neuroprotective properties of DHA. This paper demonstrates that this omega-3 PUFA can correct endolysosomal pathway defects in SH-SY5Y APPswe cells by promoting optimal membrane organization that likely prevents C99 from interacting with Rab5, which appears to be a critical neurotoxic step in the cascade leading to early endosome expansion, the blockade of endolysosomal trafficking, and reduced exosome production. These results could help to consider the adoption of an adjuvant treatment based on DHA, which could prolong the normal functioning of the aging brain and reduce susceptibility to neuronal death.

To our knowledge, this is the first report demonstrating the ability of DHA to correct a major dysfunction in neurons chronically exposed to amyloid stress, probably due to the membrane-stabilizing properties of this fatty acid. Additional research is still necessary to elucidate the molecular mechanisms linked to the effects induced by this PUFA on membrane remodeling and notably on the regulation of endosome trafficking. DHA supplementation could represent a safer and less expensive adjuvant therapeutic strategy than pharmacological treatments for AD, offering very diverse potential benefits for the numerous pathways involving membrane proteins.

## Figures and Tables

**Figure 1 ijms-25-06816-f001:**
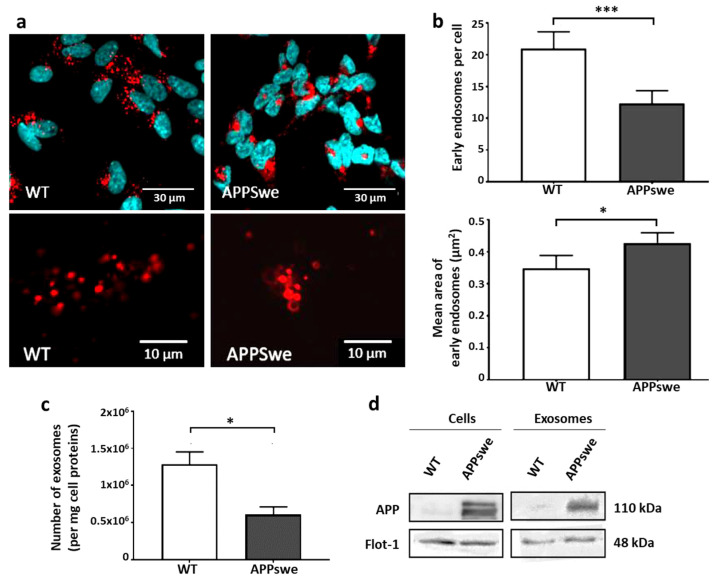
Immunochemical characterization of the endosomal alterations observed in SH-SY5Y APPswe cells. (**a**) Immunochemical staining of EEA1 (red) and nuclei (blue) in WT and APPswe SH-SY5Y cells. (**b**) Number and area of early endosomes, referring to EEA1-positive spots normalized to the cell number. Signals were detected using confocal microscopy, at 60X and 120X magnification, and quantified using ImageJ 1.52 software. Results are presented as means (±SEM, N = 4 independent experiments, n = 5 separate random fields per experiment). (**c**) NTA quantification of exosomes secreted in the culture media. (**d**) Western blot demonstration of the presence of APP specifically in the APPswe cells and their related exosomes, using flotillin-1 as a reference (analyzed from a distinct blot). Selected lanes have been cut out from the original blot images to allow for a representative profile to be shown. Results are presented as means (±SEM, N = 3 independent samples and n = 5 measurements per sample). Statistical analyses were performed via Student or Mann–Whitney tests, and significant differences are indicated as * *p* < 0.05 and *** *p* < 0.001.

**Figure 2 ijms-25-06816-f002:**
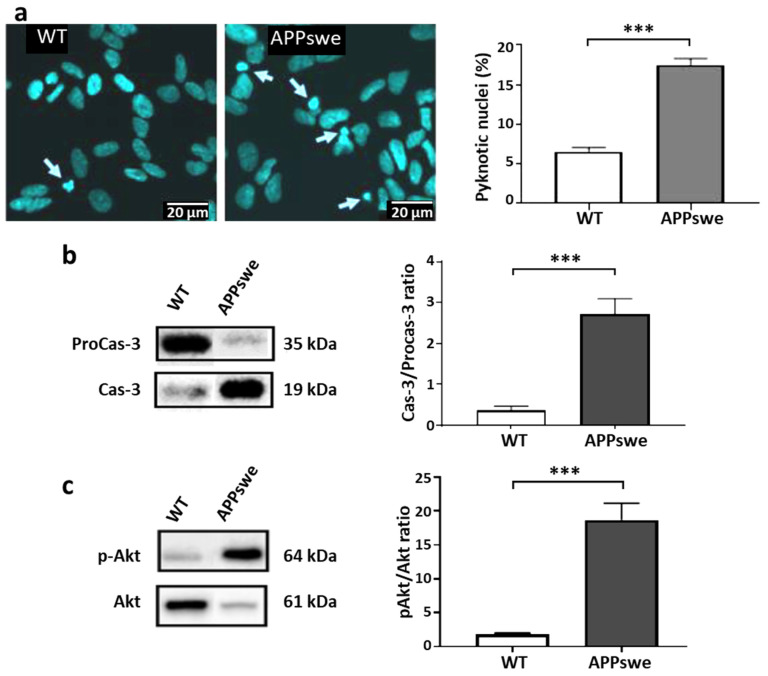
Activation of death signaling pathways in SH-SY5Y APPswe cells. (**a**) DAPI staining of WT and APPswe cells. Pyknotic nuclei are indicated with arrows. The proportion of pyknotic nuclei was calculated relative to total nuclei of the cells in the plate. Results are presented as means (±SEM, N = 4 independent experiments, n = 5 separate random fields per cell line). Statistical analyses were performed via Student or Mann–Whitney tests, and significant differences are indicated as *** *p* < 0.001. Representative immunoblot profiles (from separate blots) and corresponding to caspase-3/procaspase-3 (**b**) and p-Akt/Akt (**c**) ratios in WT and APPswe cells. Selected lanes have been cut out from the original blot images to allow a representative profile to be shown. Results are presented as means (±SEM, N = 3 independent experiments, n = 3 separate samples). Statistical analyses were performed via a Mann–Whitney test, and significant differences are indicated as *** *p* < 0.001.

**Figure 3 ijms-25-06816-f003:**
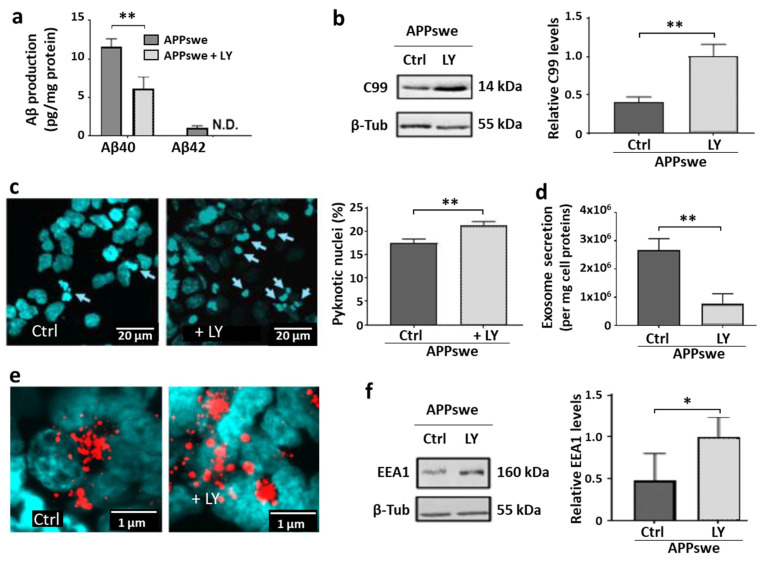
Effects of LY-mediated γ-secretase inhibition in SH-SY5Y APPswe cells. The cells were treated with 0.5 µM LY for 24 h and compared with untreated cells (Ctrl). Selected immunoblot lanes have been cut out from the original blot images to allow a representative profile to be shown. (**a**) Reduction in Aβ40 and Aβ42 production. (**b**) Accumulation of the C99 protein. Representative immunoblot profiles and densitometric analysis normalized with β-tubulin (from separate blots). (**c**) DAPI staining. Pyknotic nuclei are indicated with arrows and were counted relative to total nuclei in the cultured cells. Results are presented as means (±SEM, N = 4 independent experiments, n = 5 separate random fields). (**d**) Exosomes secreted in the medium, quantified using NTA. Results are shown as means (±SEM, N = 3 independent experiments, n = 3 different samples and 5 measurements per sample). (**e**) EEA1-labeled vesicles (red spots) studied via immunochemical staining using appropriate monoclonal antibodies. (**f**) EEA1 levels studied via Western blot analysis, quantified through densitometry, and normalized to β-tubulin (from separate blots). Data are shown as means (±SEM) of three independent experiments with n = 3 separate samples. Statistical analyses were performed via Student or Mann–Whitney tests, and significant differences are indicated as * *p* < 0.05 and ** *p* < 0.01. N.D. = not detected.

**Figure 4 ijms-25-06816-f004:**
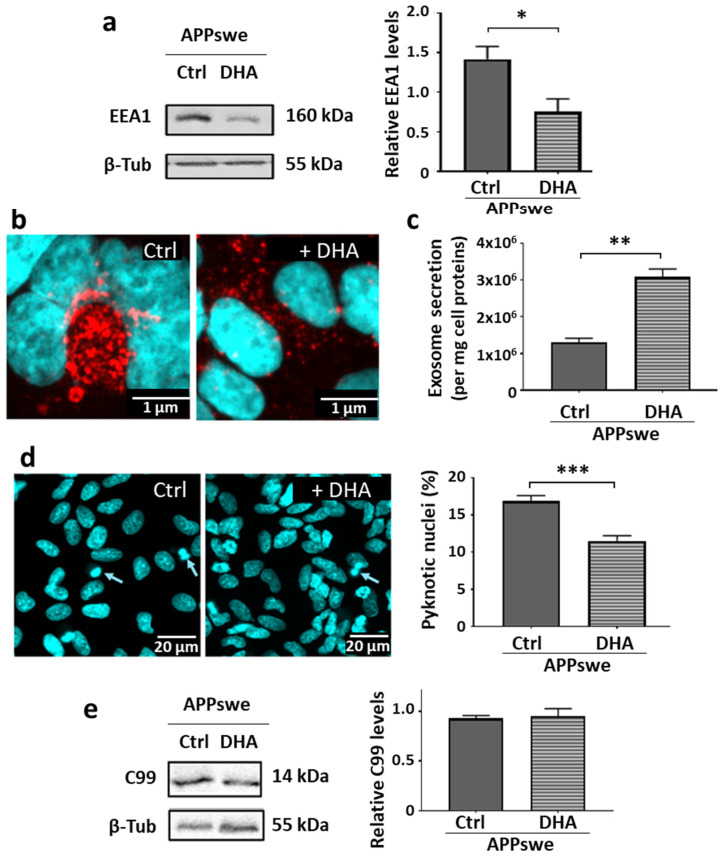
Protective effects of DHA on endogenous toxicity in SH-SY5Y APPswe cells. SH-SY5Y APPswe cells were treated or not with 0.5 µM LY for 48 h. Selected immunoblot lanes have been cut out from the original blot images to allow a representative profile to be shown. (**a**) Reduced EEA1 proteins. Representative immunoblot profiles are shown, as well as densitometric analysis after normalization with β-tubulin (from separate blots). (**b**) Size reduction of EEA1-positive vesicles (red spots). (**c**) Increased exosome production, quantified using NTA; data are shown as means (±SEM, N = 3 independent experiments, n = 3 different samples and 5 measurements per sample). (**d**) Reduction in the DAPI-stained ratio of pyknotic nuclei (indicated with arrows) relative to total nuclei of the cells in the plate. (**e**) Unchanged levels of the C99 protein. Representative immunoblot profiles are shown, as well as a densitometric analysis after normalization with β-tubulin (from separate blots). Results are presented as means (±SEM, N = 4 independent experiments, n = 5 separate random fields per condition). Statistical analyses were performed via Student and Mann–Whitney tests, and significant differences are indicated as * *p* < 0.05, ** *p* < 0.01, and *** *p* < 0.001.

**Figure 5 ijms-25-06816-f005:**
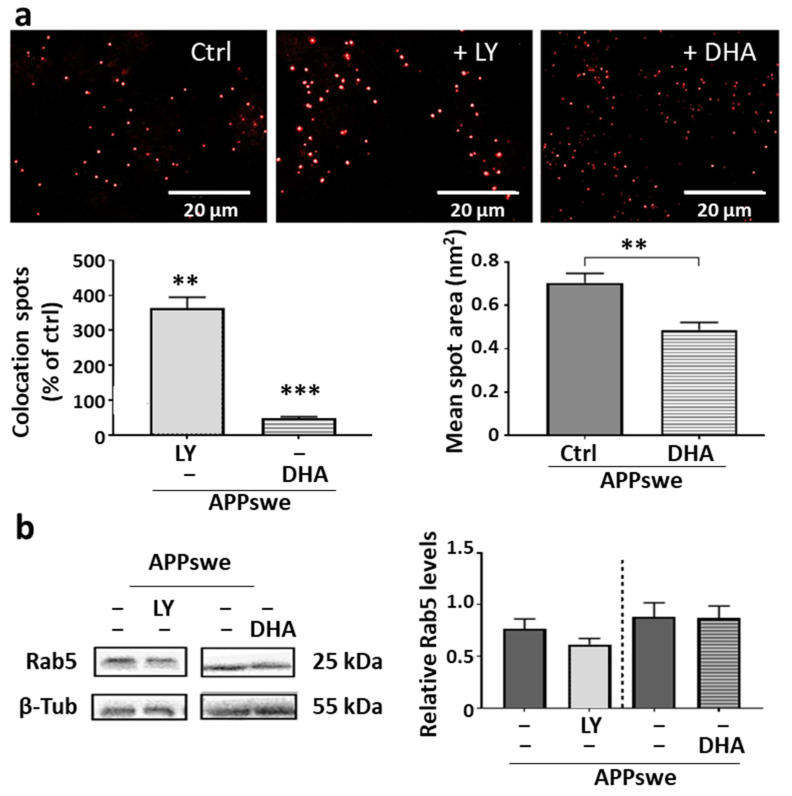
Effects of LY and DHA treatment on Rab5 protein. Selected immunoblot lanes have been cut out from the original blot images to allow a representative profile to be shown. (**a**) Effects on colocation of Rab5 and C99 proteins, studied via Duolink^®^ analysis, with colocation spots (in red) detected using fluorescence microscopy and quantified with Image J. (**b**) Effects on Rab5 levels; representative profiles are shown, as well as a densitometric analysis after normalization with β-tubulin (from separate blots). Data are shown as means (±SEM), N = 2 experiments, n = 3 samples and 5 measurements per sample). Statistical analyses were performed via Student and Mann–Whitney tests, and significant differences are indicated as ** *p* < 0.01 and *** *p* < 0.001.

## Data Availability

The data presented in this study are publicly available from the following link: https://doi.org/10.57745/EKMKOX.

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
