# Peer review of "Increasing the Survival of a Neuronal Model of Alzheimer’s Disease Using Docosahexaenoic Acid, Restoring Endolysosomal Functioning by Modifying the Interactions between the Membrane Proteins C99 and Rab5"

_ijms, 2024, doi:10.3390/ijms25136816_

Round 1
Reviewer 1 Report (Previous Reviewer 3)
Comments and Suggestions for Authors
I can only sincerely congratulate the authors for presenting such a well-structured and well-done piece of work.
Author Response
We are pleased to have been able to meet the reviewers' recommendations and we humbly thank them for suggesting constructive and positive improvements.
Reviewer 2 Report (New Reviewer)
Comments and Suggestions for Authors
The authors are using retinoic acid treated SH-SY5Y human neuroblastoma cells as model for Alzheimer’s disease to investigate the role of docosahexaenoic acid (DHA) on the dysfunction of the endo-lysosomal pathway. Comparison between wild type cells and those expressing the Swedish form of APP precursor protein is presented. Persistent expression of the mutated C99 fragment causes endosome enlargement, reduced secretion of exosomes and higher levels of apoptosis. Neuroprotection by DHA treatment was revealed by decreased numbers of pyknotic nuclei which was associated with a reduced interaction of Rab5GTPase with C99.
Major Point: Investigations in live animal models are not shown to verify the data obtained here in 2D cell culture, exclusively.
Minor points:
In section 2.5, line 240, a “Duolink” colocalization assay is mentioned but the method of proximity ligation assay was not mentioned otherwise although, data in Figure 5 are crucial for the conclusion of this manuscript. The "Duolink" method should be explained including negative controls to verify the specificity of interaction, in the Figure legend or in the methods section.
There is a recent paper by D`Aloia et al describing the cellular model of functional cholinergic-like neurons developed by reprogramming the human SH-SY5Y neuroblastoma cell line. (2024); https://doi.org/10.1038/s41420-023-01790-7. This paper could be cited and critically discussed including a cautionary statement concerning the investigation of DHA in 3D cultures of SH-SY5Y neuroblastoma cells, in primary cholinergic neurons or in the brain.
Author Response
- Major point. Obviously, we strongly defend the results and conclusions contained in our manuscript, but as Reviewer 2 must have experienced himself, several steps are still necessary before moving from this work on cellular models to in vivo studies to give ourselves the best chance of answering our scientific questions on this subject.
- Minor point 1. The absence of any non-specific signal was verified in various negative controls performed with PLA probes either used alone or used with each antibody . In all these controls, no signal was observed.
- Minor point 2. We thank Reviewer 2 for pointing us to the paper by D'Aloia et al. Presenting a 3D culture model of SH-SY5Y neuroblastoma cells, this paper indeed offers the prospect of an alternative tool to 2D culture studies that can also avoid the need for animal model studies. As requested, this paper has been discussed in this context and referenced in the new revised version.
This manuscript is a resubmission of an earlier submission. The following is a list of the peer review reports and author responses from that submission.
Round 1
Reviewer 1 Report
Comments and Suggestions for Authors
Reviewer comments and suggestions
The authors in this study evaluated whether neuroprotection by Docosahexaenoic acid (DHA) can also preserve the endolysosomal function. AD-typical endolysosomal abnormalities were recorded in differentiated human SH-SY5Y neuroblastoma cells expressing the Swedish form of human APP. The characteristics included endosome enlargement, reduced secretion of exosomes and higher level of apoptosis, which verified the relevance of the cellular model chosen for studying the associated deleterious ways. The second proposed mechanism highlighted neuroprotection by DHA was associated with reduced interaction of C99 with the Rab5 GTPase, lower endosome size, restored exosome production, and lowered neuronal apoptosis. Hence the finding suggests that DHA may affect the location and interactions of proteins in the membrane environment of the neuron, correcting endocytosis and vesicular trafficking dysfunction linked to AD.
Overall, the manuscript was good. However, a few major concerns/comments needed to be explained or modified.
- Line 20 it should be “AD”
- Line 24 first time used APP, so it will be in full form.
- Line 26 Please explore it in a better way
- Line 36-39 The lines need appropriate references.
- Line 45 is this correct word “participant.”
- Line 57 Please explain the studies rather than citing only
- Line 88 What does it mean based on previous work, the authors can write few points and then cite them here
- Line 94-97 no need to mention
- Line 164, see 32 for review is the correct word to use it
- Line 214-215 what does it mean, please explain it
- Line 252-255 First paragraph should not be a general sentence, please mention the novelty of this study
- Please mention the figure result in the text of discussion for easy go-through of the manuscript
- Line 318-319 What would be the possible reason for this observation?
- Line 345-346 Several mechanisms were written by authors but did not cite any appropriate references, better to add relevant one
- Please clarify the statement that authors want to present here “even more when the activity of γ-secretase is inhibited, their interactions with Rab5 are favored and increase the occurrence of this blockage.”
- Line 369-370 Why the activity of DHA was discussed at the end?
- Line 375 a typo error was present in the line
- Line 377-379 The authors should write 2-3 references to validate the points for studies in the literature.
Reviewer 2 Report
Comments and Suggestions for Authors
In the article titled: “Increasing the survival of a neuronal model of Alzheimer's disease by docosahexaenoic acid, restoring endolysosomal functioning by modifying the interactions between the membrane proteins C99 and Rab5” is investigating the neuroprotective role of DHA by preserving endolysosomal function.
As a vitro model for AD was used differentiated human SH-SY5Y neuroblastoma cells expressing the Swedish form of human APP which revealed altered phenotype included endosome enlargement, reduced secretion of exosomes and higher level of apoptosis. The authors proved that neuroprotection by DHA was associated with reduced interaction of C99 with the Rab5 GTPase, lower endosome size, restored exosome production, and reduced neuronal apoptosis.
Introduction is well written and focused on the studied mechanisms.
Results are well illustrated with high quality figures and graphs
Discussion is in-depth and proves the hypothesis put forward at the beginning of the article about the importance of endosomal trafficking in the pathogenesis of Alzheimer's and the potential role of DHA for restoring the endosomal pathway.
My opinion is that the article can be published in the present form.
Reviewer 3 Report
Comments and Suggestions for Authors
The authors investigated the effect of DHA supplementation on endosomal trafficking and exosomal production in an AD cell model (human neuroblastoma SH-SY5Y) expressing recombinant human APP protein with the Swedish double- mutation K670N/M671L (APPswe), which promotes amyloidogenic processing of APP. The results suggest that DHA protects neurons from apoptosis by unblocking vesicular trafficking and restoring the endolysosomal pathway.
It is a very interesting work. It is very well written, and I read it with a great pleasure. The design of the experiments is clear and logical. The demonstrations are very elegant and clear. The description of the results is precise. The discussion is very engaging, well presented and as a cell neurobiologist I found it extremely interesting. As for my observations on the paper, I can only sincerely congratulate the authors for presenting such a well-structured and well-done piece of work. I found only three very minor details that I would like to point out.
MINOR
The legends to the illustrations should include the meaning of all the letters mentioned in the plates (as in Figure 1 APP, Flot-1, APPswe, Wt, etc.), but this may depend on the style of the journal.
Line 132. mention the full name of Flot-1 (Flotillin) and also include it in the legend of Figure 1.
Line 375, “Therefore” in lower case
Reviewer 4 Report
Comments and Suggestions for Authors
Comments:
The Introduction and Methods sections give sufficient background information.
The conclusions of the paper depend significantly on Western blot analyses and quantifications. Therefore careful evaluation of the Western blot results is required (which is also important because blots were developed using enhanced chemiluminescence which can be problematic for quantifications). However, selection and presentation of the blots shown in the main manuscript (Figures 1 to 5) are problematic:
First, no molecular mass markers are shown (neither in the main manuscript nor in the „original images“ file); the molecular mass indicated in the Figures are likely the expected mass of the examined proteins; but because of the lack of the marker bands, the reader is unable to judge the correct mass; which would be especially importnat e.g. in case of APP (Figure 1c) were several bands are stained (from the labeling it is diffcult to conclude which of the bands is the 110 kDa band; are all bands specific?).
Second, the Western blots shown in Figures 1-5 (actually only single bands are shown) are composed of different lanes of the original blots. In some Figures several bands were cut out; this is obvious when examing the „original images“, however this manipulation this is not easily visible in the Figures and should be indicated in a more conspicuous way.
Third, the number of WT and APPswe samples in the original images of the blots differs considerably; in some cases the sample numbers in the blot of the examined protein and in the control protein are different, although one should expect that the same samples were analyzed; normally the control protein should be stained on the same blot (to serve as loading control and to ensure that the same samples were stained for the examined protein and the control). For example: In Figure 2b and 2c, Western blot data are, according to the Figure legend representative of 3 independent experiments and n=3 separate samples: it is not clear are these 3 independent experiments each with 1 sample? or have there been 3-times 3 = 9 samples per cell type? The original blots show 6 WT samples for ProCas-3 and 6 for APPswe, but 5 samples WT for Cs-3 and 7 for APPswe, however, the same samples should have been examined with Procas-3 and Cas-3 antibodies; and the same blots should have been developed with ProCas-3 and Cas-3 antibodies sequentially and the results with ProCas-3 and Cas-3 obtained with the same samples should be shown in the article. It is difficult to believe that for the ProCas-3 Western blot 6 WT and 6 APPswe samples were examined and for the corresponding blot of total caspase-3 only 5 WT, but 7 APPswe were analyzed.
Similarly, P-Akt and Akt Western blots show different number of samples.
Figure 3b (C99 / tubulin) and 3f; again C99 or EEA1 and tubulin signals are from different blots; however, the tubulin blot should be the loading control (and should be from the same blot as the C99 and EEA1 signals);
Minor points:
Figure 1C: „2.106“ and so on should be something like „2x106“ (also in the other Figures), especially because the period is also used in „1.5“